# Hybrid Immunity Improves the Immune Response after the Fourth COVID-19 Vaccine Dose in Individuals with Medical Conditions Predisposing to Severe COVID-19

**DOI:** 10.3390/vaccines12030247

**Published:** 2024-02-27

**Authors:** Nina Ekström, Tuija M. Leino, Aapo Juutinen, Toni Lehtonen, Anu Haveri, Oona Liedes, Saimi Vara, Heini Salo, Arto A. Palmu, Hanna Nohynek, Timi Martelius, Merit Melin

**Affiliations:** 1Department of Health Security, Finnish Institute for Health and Welfare, 00271 Helsinki, Finland; tuija.leino@thl.fi (T.M.L.); aapo.juutinen@thl.fi (A.J.); anu.haveri@thl.fi (A.H.); oona.liedes@thl.fi (O.L.); saimi.vara@thl.fi (S.V.); heini.salo@thl.fi (H.S.); hanna.nohynek@thl.fi (H.N.); merit.melin@thl.fi (M.M.); 2Department of Knowledge Brokers, Finnish Institute for Health and Welfare, 00271 Helsinki, Finland; toni.lehtonen@thl.fi; 3Department of Public Health and Welfare, Finnish Institute for Health and Welfare, 33100 Tampere, Finland; arto.palmu@thl.fi; 4Division of Infectious Diseases, Inflammation Center, Helsinki University Hospital, 00290 Helsinki, Finland; timi.martelius@hus.fi

**Keywords:** COVID-19, SARS-CoV-2, immunocompromised, COVID-19 vaccine, immune response, booster, hybrid immunity, neutralization

## Abstract

Data on immune responses following COVID-19 booster vaccinations and subsequent infections in the immunocompromised are limited. We studied antibody responses after the fourth dose and subsequent infections to define patient groups benefiting most from boosters. Fourth vaccine (booster) doses were, in Finland, first recommended for severely immunocompromised individuals, whom we invited to participate in our study in 2022. We assessed spike protein-specific IgG and neutralizing antibodies (NAb) against the ancestral and Omicron BA.1 strains one month after the fourth dose from 488 adult participants and compared them to the levels of 35 healthy controls after three doses. We used Bayesian generalized linear modeling to assess factors explaining antibody levels and assessed vaccine-induced and hybrid immunity six months after the last vaccine dose. Chronic kidney disease (CKD) and immunosuppressive therapy (IT) were identified as factors explaining sub-optimal antibody responses. The proportion of participants with a normal antibody response and NAbs was significantly lower regarding CKD patients compared to the controls. By the 6-month sampling point, one-third of the participants became infected (documented by serology and/or molecular tests), which notably enhanced antibody levels in most immunocompromised participants. Impaired antibody responses, especially NAbs against the Omicron lineage, suggest limited protection in individuals with CKD and highlight the need for alternative pharmaceutical preventive strategies. Vaccination strategies should take into account the development of robust hybrid immunity responses also among the immunocompromised.

## 1. Introduction

Immunocompromised individuals are at an increased risk for severe coronavirus disease 2019 (COVID-19) and at a higher risk of mortality after hospitalization due to COVID-19 compared to immunocompetent individuals [1,2]. The risk depends heavily on the extent to which the immune system has been suppressed [3]. Although vaccinations aim to protect these most vulnerable groups from severe COVID-19, immunocompromised individuals have been shown to exhibit impaired responses to vaccines, including impaired or delayed antibody production [4,5], diminished T-cell responses in some studies [6,7], and greater waning of immunity leading to waning vaccine effectiveness [8,9], and they may not achieve the same level of protection as the immunocompetent [10,11]. This may be due to the immunomodulatory effects of the disease or condition itself or due to treatment with immunosuppressive effects.

A primary series of two COVID-19 vaccine doses in immunocompromised populations generates impaired immune responses compared to the general population [12]. The World Health Organization (WHO) therefore recommended a third dose for all immunocompromised persons [13]. In Finland, a fourth COVID-19 vaccine dose was recommended at a minimum interval of three months after the third dose in populations at risk for severe COVID-19 in late 2021 [14]. At the same time, a third (booster) dose was recommended for healthcare workers, older residents in long-term care facilities, and subsequently for all adults [15]. Thereafter, in 2022–2023, subsequent booster doses have been recommended for those at greatest risk for severe COVID-19. However, specific information on which patient groups would benefit most from booster doses remains limited.

With the spread of the Omicron variants of severe acute respiratory syndrome coronavirus 2 (SARS-CoV-2), vaccinations have provided only very limited and short-lived protection from COVID-19 infections, even in the boosted immunocompetent population. We have estimated that 39% of the Finnish adult population had developed antibody-mediated hybrid immunity induced by a combination of vaccinations and infections during the first half of 2022 and over half by the end of 2022 [16]. However, data on the development of hybrid immunity in different immunocompromised disease groups are scarce. With the continued evolution of new SARS-CoV-2 variants with increasing potential to evade immunity [17,18], data on the persistence of vaccine-induced and hybrid immunity are required to inform vaccination strategies in this high-risk population.

In this study, we examined antibody levels after the fourth COVID-19 vaccine dose in a cohort of severely immunocompromised adults with varying underlying conditions. We aimed to identify patient groups who could benefit from additional booster vaccinations the most. We used Bayesian modeling for the evaluation of factors explaining diminished antibody responses. As the fourth vaccine dose for the immunocompromised individuals was considered their first booster dose, we compared their IgG and neutralizing antibody levels against the ancestral SARS-CoV-2 strain and Omicron BA.1 variant to levels in healthy controls after the third dose (booster). We additionally evaluated the persistence of vaccine and hybrid immunity-induced antibodies up to six months after the fourth dose in immunocompromised individuals with different underlying conditions.

## 2. Materials and Methods

### 2.1. Study Design and Participants

The COVID-19 vaccine immunological study is a clinical trial conducted in Finland by the Finnish Institute for Health and Welfare (THL) within the well-being services counties of Uusimaa, Pirkanmaa, Kanta-Häme, Central Finland, Päijät-Häme and Ostrobothnia. No investigational products were used. The participants received COVID-19 vaccines, in accordance with the national recommendations, through routine healthcare services. We included two main study cohorts in the study: immunocompromised individuals and healthy controls (Figure 1).

For individuals with severe predisposing medical factors affecting the risk of COVID-19, a third vaccine dose was recommended (on 17 September 2021) to supplement the primary vaccination series at a minimum of two months after the second vaccine dose, but due to the short interval, this dose was not considered a booster [19]. For this group, the recommendation of a fourth COVID-19 vaccine dose at a minimum of three months following the previous dose was given on 22 December 2021 [20], and it was considered the first booster dose.

In Finland, a third vaccine dose was recommended (on 21 September 2021) at a minimum of six months after the primary vaccination series for healthcare workers and the elderly who had received the primary series doses with a short interval (<6 weeks) [21]. Later, on 26 November 2021, third vaccine doses were recommended for all healthcare workers at a minimum of six months following the second dose [22].

For the study cohort of immunocompromised individuals, we invited 1440 subjects (aged ≥ 18 years) via random sampling in January 2022. The inclusion criteria were a previous receipt of three doses of a monovalent COVID-19 vaccine (BNT162b2, mRNA-1273 or ChAdOx1 nCoV-19) followed by a fourth monovalent vaccine dose (BNT162b2 or mRNA-1273) by 25 January 2022. A documented previous SARS-CoV-2 infection was an exclusion criterion. The subjects were invited via regular mail and were asked to donate a blood sample at their local healthcare district laboratory after the fourth dose. Any previous laboratory-confirmed SARS-CoV-2 infections were verified from register data. All study subjects were also asked, on their first study visit, whether they had previously received a positive result from an at-home antigen test. Seropositivity with a nucleoprotein IgG antibody test was additionally used to identify previous infections. Study subjects with a known previous SARS-CoV-2 infection were excluded from this study (*n* = 58).

To evaluate a normal vaccine response, we invited 65 healthcare workers from the Helsinki University Hospital (HUS) in December 2021. Of them, 35/65 had no predisposing medical factors affecting the risk of COVID-19 (as confirmed from healthcare registers) and no known previous SARS-CoV-2 infections, and these were selected as the healthy control group for this study (Figure 1). They received their first two vaccine doses (BNT162b2 or ChAdOx1 nCoV-19) at a dose interval of >6 weeks (median 84, IQR 83–85 days), and they received the third, i.e., the booster dose (BNT162b2), 210 days (median, IQR 198–234) after the second dose and provided serum samples at 36 days (median, IQR 31–42) after the third dose (BNT162b2) in January–February 2022. Their median age was 45 years (IQR 37–56), and 86% were female.

We subsequently invited all study participants to a follow-up visit six months after the fourth (median 5.6 months, IQR 5.4–5.9) or third dose (i.e, healthy control group; median 6.0 months, IQR 5.7–6.2) in May–July 2022 (Figure 1). Of the immunocompromised participants, 109/356 (31%) and 5/14 (36%) of the participants in the control group experienced SARS-CoV-2 infection during the follow-up and provided post-infection samples for assessments of hybrid immunity (Figure 1). The participants were considered to have hybrid immunity if they had been infected with SARS-CoV-2 between the first and second study visit (at least seven days before the second sampling).

A small number (*n* = 38) of the participants who had received four doses of the COVID-19 vaccine at the time when the fourth dose was recommended only to severely immunocompromised individuals had no registered predisposing medical factors affecting the risk of COVID-19. These participants were included in the study as an additional control group, which enabled us to the assess antibody responses after three and four doses in participants with no known predisposing medical factors.

We retrieved information on COVID-19 vaccinations from the National Vaccination Register (THL), information on laboratory-confirmed SARS-CoV-2 infections from the National Infectious Diseases Register (THL), and information on predisposing factors for COVID-19 from the Care Register for Health Care (THL), the Register of Primary Health Care Visits (THL), the Special Reimbursement Register for Medicine Expenses and the Prescription Center database (the Social Insurance Institution of Finland, KELA, Helsinki, Finland).

### 2.2. Laboratory Analyses

#### 2.2.1. SARS-CoV-2 Fluorescent Multiplex Immunoassay

We measured the concentration of serum IgG antibodies with an in-house fluorescent multiplex immunoassay (FMIA) [23] to the SARS-CoV-2 nucleoprotein (N-IgG) and two spike protein (S-IgG) antigens: full-length spike protein (SFL) and the receptor binding domain of spike protein (RBD). We used a threshold of 10 binding antibody units (BAU)/mL for N-IgG seropositivity, resulting in 93% specificity and 100% sensitivity, and thresholds 2 and 3 BAU/mL for SFL-IgG and RBD-IgG, with 100% sensitivity and specificity, respectively. The assay was calibrated to the WHO international standard [24].

#### 2.2.2. Microneutralization Assay

A live virus cytopathic effect-based a microneutralization assay (MNT) [25,26] was performed to determine neutralizing antibody (NAb) titers against SARS-CoV-2 in subgroups of randomly selected samples (*n* = 17–20/risk group). NAb titers were additionally determined for the first 27 samples taken of the 35 healthy controls. We used two SARS-CoV-2 viruses isolated in Finland, between 2020 and 2022, representing the ancestral strain (WT) and the Omicron BA.1 subvariant [27]. Omicron BA.1 was the most prevalent variant in Finland from the end of 2021 to March 2022, followed gradually by Omicron BA.2 [28]. The isolation and propagation of the WT strain were performed in African green monkey kidney epithelial (Vero E6) cells [25] and the BA.1 strain in VeroE6-TMPRSS2-H10 cells [29] and further propagated in Vero E6 cells for MNT. A tissue culture infectious dose 50% assay was performed for both viruses to achieve a comparable virus concentration among the different strains. Results were expressed as MNT titers; MNT titer ≥ 6 was considered positive, borderline positive when 4, and negative when <4. Borderline positive values were further confirmed by biological repeats.

### 2.3. Statistical Methods

We assessed IgG concentrations in the control group one month after the third dose and used the 5th percentile as the threshold for a normal response. To gain a more comprehensive insight into the variability and distribution of the normal response, we employed bootstrapping, generating 50,000 samples. This allowed us to estimate also the 95% highest density interval (HDI), providing a robust characterization of the response’s uncertainty and distribution.

Since most study participants had more than one immunosuppressive condition, a direct comparison of individual patient groups was not possible in this study setting. We therefore used modeling to help identify diseases and other factors that would explain low IgG concentrations one month after the fourth vaccine dose. We utilized a Bayesian gamma regression model with IgG concentrations as the response variable and age, sex, the fourth vaccine product, and the risk group status for various predisposing conditions as explanatory variables. Assuming that only some of the explanatory variables truly impact the IgG concentrations, we used horseshoe priors [30] for the regression coefficients to induce sparsity in the model estimates. The model was applied separately for RBD- and SFL-IgG concentrations. The reference levels for categorical variables were defined as male for sex, BNT162b2 for the fourth vaccine product and no risk factors for the risk group.

We calculated the geometric mean antibody concentrations (GMCs) and titers (GMTs) with 95% confidence intervals (CIs) for IgG and NAb levels, respectively, and compared the concentrations and titers of each subgroup to the control group and the differences in antibody concentrations between vaccine and hybrid immunity groups via the Mann–Whitney test. MNT titers < 4 were assigned a value of 2. We compared the proportions of subjects with a normal antibody concentration (RBD-IgG ≥ 398 BAU/mL and SFL-IgG ≥ 716 BAU/mL), MNT titer ≥ 4 and those who received mRNA-1273 as the fourth dose with Fisher’s exact test. Wilcoxon signed-rank test was used to compare antibody levels between different time points within groups. We calculated the Spearman correlation coefficient (ρ) and the statistical significance of the correlation between antibody concentrations and age and between antibody concentrations and MNT titers. The statistical significance level of difference was set to *p* < 0.05 and adjusted with Bonferroni correction to *p* < 0.0025–0.025, depending on the number of groups, time points, and antibody/strain specificities in the comparison. Statistical analyses were performed with GraphPad v9 and R version 4.3.2.

## 3. Results

### 3.1. Characteristics of the Participants Who Received Four Vaccine Doses

In all, 602/1440 (42%) invited subjects participated (Figure 1). Of them, 488 (81%) had received their first two vaccines (BNT162b2, ChAdOx1 nCoV-19 or mRNA-1273) with a dose interval of >6 weeks (IQR 83–84 days) followed by a third dose (BNT162b2, ChAdOx1 nCoV-19 or mRNA-1273) 112 days (median, IQR 90–128) after the second dose. They received their fourth dose 106 days (median, IQR 99–112) after the third dose and provided their first sample for the study 36 days (median, IQR 32–41) after the fourth vaccine dose (BNT162b2 or mRNA-1273). We included these 488 participants in this study (Figure 1, Table 1). Half of the participants (264/488; 54%) received BNT162b2, and the other half (224/488, 46%) received mRNA-1273 as the fourth dose. The median age was 64 years (IQR 52–69), and 58.4% were female (Table 1). The majority of the participants (394/488; 81%) had more than one (IQR 2–4) medical condition predisposing to severe COVID-19 (Table 2, Appendix A). 

### 3.2. Impact of Different Factors on Antibody Levels after the Fourth Dose

Since most participants who had received the fourth vaccine dose had more than one medical condition predisposing to severe COVID-19 and the patient groups largely overlapped, we were not able to directly compare antibody responses between different patient groups in this study setting. We used the Bayesian gamma regression model to assess which factors best explain low antibody concentrations after the fourth dose. The results presented in Table 2 show the lowest estimates of posterior means for regression coefficients explaining SFL- and RBD-IgG antibodies for chronic kidney disease (CKD) (0.479 [0.339–0.741]; 0.490 [0.342–0.787]) and immunosuppressive therapy (IT) (0.612 [0.455–0.845]; 0.652 [0.482–0.940]), respectively.

Based on the modeling data, the effects of sex or age on antibody levels following the fourth dose were not significant. mRNA-1273 as the fourth dose (irrespective of the previous vaccine products given) resulted in the highest estimates (1.568 [1.161–2.018]; 1.510 [1.092–1.968]) suggesting better immunogenicity than that of BNT162b2 (Table 2). However, since there was no information recorded in the National Vaccination Register on whether mRNA-1273 was given as a whole or half dose, we could not reliably evaluate the immunogenicity of the mRNA-1273 compared to BNT162b2 when given as a booster vaccine. The recommendation to halve the vaccine dose when mRNA-1273 was used as a booster was given in October 2021 [31]. However, it is possible that, especially for those groups where the vaccine response was expected to be weaker, the full dose continued to be used. However, since mRNA-1273, given as the fourth dose, was associated with higher antibody levels compared to BNT162b2, and participants in the control group had not received mRNA-1273, we included only those immunocompromised participants who had received BNT162b2 as their fourth dose in further analyses (Figure 1).

In this study, we focused further evaluation of IgG and neutralizing antibody levels on the group of CDK patients and those receiving IT. As with most participants, many of the participants living with CDK had comorbidities that affect immunocompetence. A total of 22/29 (76%) of participants living with CKD were organ transplant (OT) recipients. In this study, we evaluated antibody levels in participants living with CKD or CKD with organ transplant (CDK with OT) but not CKD alone (only seven subjects; Appendix A). We did not include the group with OT (without CKD) in the assessment of NAbs because of limited register data available of the time and type of the transplantation.

The IT group was large and heterogeneous, i.e., immunosuppressive treatment was combined with various diseases that may have also weakened the immune response. Primarily, we would have selected participants receiving biological drugs (e.g., anti-CD20 therapies such as Rituximab) which are expected to affect antibody-mediated immune responses. However, as these drugs are administered in hospitals, the information is not recorded in the registers to which we had access in this study (the Special Reimbursement Register for Medicine Expenses and the Prescription Centre database, the Social Insurance Institution of Finland, KELA). Since participants living with rheumatoid diseases (RDs) formed a large group within the IT group, and biological drugs (including Rituximab) are used in the treatment of RD, we selected participants of this group to represent those receiving IT treatment (Figure 1).

### 3.3. Antibody Levels and NAbs after Third Dose in Healthy Controls

The antibody levels in the healthy controls ranged between 338 and 4435 BAU/mL for RBD-IgG and 373 and 7036 BAU/mL for SFL-IgG, and the mean concentrations were 1414 BAU/mL (95% CI 1120–1784) and 2168 BAU/mL (95% CI 1713–2744), respectively (Figure 2). We assigned the lower limit of a normal antibody response to vaccination at the fifth percentile of the antibody concentrations: 398 and 716 BAU/mL for RBD-IgG and SFL-IgG, respectively. All controls had NAb against WT and Omicron BA.1, with GMTs of 404 (95% CI 269–608) and 26 (95% CI 18–35), respectively (Figure 3).

### 3.4. Antibody Levels and NAbs One Month after the Fourth Dose by Disease Groups

Here, we assessed the IgG antibody levels in the immunocompromised participants living with (1) CKD (with or without OT), (2) CKD with OT or (3) RD (with IT). We assessed the NAb levels in a small subgroup of participants living with (1) CKD (with or without OT), (2) CKD with OT and (3) those who received IT (without CKD and with or without RD).

The majority (93%) of immunocompromised participants were seropositive for S-IgG antibodies after the fourth dose (Figure 2). The percentage of participants with S-IgG antibody levels exceeding the lower limit of a normal response was significantly lower in the CKD and CKD with OT groups (31 and 18%) compared to the healthy controls (*p* < 0.001) (Table 3). In the CKD and CKD with OT groups, the S-IgG antibody concentrations were significantly lower compared to the control group (*p* < 0.0025; Figure 2). Within the CKD and CKD with OT groups, SFL and RBD concentrations significantly correlated negatively with age (ρ= −0.438–−0.526, *p* < 0.025), except RBD-IgG in the CKD with OT group (Figure 4).

We evaluated neutralizing antibodies for a randomly selected subgroup of participants with CKD (*n* = 20, of whom with OT, *n* = 16) or IT (without CKD) (*n* = 20, of whom with RD, *n* = 10). In the CKD, CKD with OT, and IT groups, 70%, 63%, and 90% of subjects had NAb against WT SARS-CoV-2 compared to 100% in the control group (Figure 3, Table 3). Mean NAb titers against BA.1 were >10-fold lower compared to WT in all groups. In the CKD, CKD with OT, and IT groups, 45%, 38%, and 60% of participants had NAbs against BA.1 which was significantly less (*p* < 0.004) than in the control group (100%) (Table 3). In the CKD and CKD with OT groups, the mean NAb titers against BA.1 were significantly lower compared to the control group (*p* < 0.001; Figure 3). Levels of NAbs against the WT and BA.1 strains correlated with the levels of SFL- and RBD-IgG antibodies (ρ = 0.448–0.934; Appendix A). However, the minimum RBD-IgG antibody concentration generally required to neutralize the BA.1 variant was 200 BAU/mL in the CKD and IT groups. For samples with RBD-IgG concentrations below this, the NAb titer remained negative. We estimated the percentage of participants with NAbs against the BA.1 variant in different study subgroups using this tentative cut-off of 200 BAU/mL (Table 3). In the control group, all samples had RBD-IgG concentrations ≥ 338 BAU/mL and had NAbs against the BA.1 variant (Appendix A).

In the additional control group of participants with four vaccine doses, but no registered medical factors affecting the risk of COVID-19, the mean NAb titers were comparable to titers in the control group with three doses of vaccine (Figure 3).

### 3.5. Antibody Levels Six Months after the Booster Dose in Participants with and without Infection

During follow-up, 109/356 (31%) of the immunocompromised participants experienced SARS-CoV-2 infection (Table 1). Of the infections, 68 (62%) were documented in the Infectious Diseases Register, and 42 (38%) were identified solely by a >30% increase in N-IgG and/or S-IgG concentrations. Seven participants had hospital-treated COVID-19 disease, and six of them had CKD (including four with CKD and OT), and one had cancer and severe heart disease. The median time from laboratory-confirmed SARS-CoV-2 infection to 6-month sampling was 76 days (range 12–181 days). Since the proportion of participants who received mRNA-1273 (any dose) was not significantly different in participants with or without infection by six months after the fourth dose (44 vs. 71%, 44 vs. 65%, 52 vs. 52%, and 56 vs. 50% for subjects with CKD, CKD with OT, RD, and in the additional control group (with four doses), respectively), we did not exclude mRNA-1273 recipients from the comparison of vaccine-induced and hybrid immunity. Only 14 of 35 participants in the control group (with three doses) participated in the 6-month sampling, and 5/14 had been infected following the third vaccine dose (36%). None of the participants who had an infection in the control group had been hospital-treated. The percentage of participants who had experienced SARS-CoV-2 infection was statistically not different in the immunocompromised and control group (36 vs. 31%; *p* = 0.769).

In participants without infection, the mean IgG concentrations decreased significantly in all groups; 2.0–3.2-fold in the CKD and CKD with OT, 5.8–6.1-fold in the RD, and 4.9–6.0-fold in the control group (Figure 5). The S-IgG concentrations were significantly lower in the CKD and CKD with OT groups than in the control group (*p* < 0.0025, Figure 2).

After infection, the mean S-IgG concentration increased 4.7–9.0-fold in CKD and CKD with OT, 3.7–4.2-fold in RD, and 1.9–2.5-fold in the control group. The increase in antibody concentration (before vs. after infection) was significant (*p* < 0.0001) in the RD group (Figure 5). Infection induced increases in RBD- and SFL-IgG concentrations in most subjects with CKD (68 and 79%), CKD with OT (69 and 81%), or RD (84 and 84%), respectively (Figure 5). In the control group (with three does), the RBD- and SFL-IgG concentrations increased after infection in three of the five participants. Six months after the fourth vaccine dose, RBD- and SFL-IgG concentrations were significantly higher in participants who had an infection (i.e., those who had developed hybrid immunity during the follow-up) compared to those without infection (*p* < 0.005), except for the CKD with OT group and RBD-IgG in the CKD group (Figure 5).

## 4. Discussion

In this study, we showed that the magnitude of the antibody response after the fourth vaccine dose was sub-optimal, particularly in participants with severe CKD. On the other hand, we showed that infections enhanced antibody levels in most immunocompromised participants. With the emergence of the Omicron variant and the variants that followed, infections have been widespread also in the vaccinated population. Hybrid immunity has been found to provide higher and more durable protection against reinfection and severe disease than vaccination alone [32]. The data on the immune response to the fourth dose/booster doses and hybrid immunity are limited, especially in disease-specific subgroups, e.g., individuals living with CKD or RD. Our study showed that after the fourth dose, the proportion of participants with a normal antibody response and those with NAb against the Omicron BA.1 variant was significantly lower in CKD and RD groups compared to the healthy controls. Numerous studies have consistently shown that the presence of neutralizing antibodies is a strong indicator of immune protection against SARS-CoV-2 infection [33]. Consequently, an impaired neutralizing antibody response suggests limited protection. We additionally showed that despite impaired responses to the booster vaccination, the infections boosted the antibody concentrations from what had been achieved with the fourth vaccine dose in the majority of the immunocompromised participants. Further, we conclude that mRNA-1273, as the fourth dose, was associated with higher immunogenicity compared to BNT162b2 in immunocompromised subjects.

CKD and RD associated with immunosuppressive therapy have also, in previous studies, shown to impact immune responses to COVID-19 vaccinations. Previous studies have shown overall seroconversion rates being as high as in healthy controls after the primary vaccination [34,35,36] but with significantly lower mean RBD-IgG antibody levels, NAb antibody titers (against wt), and lower T-cell responses compared to healthy subjects [6,34,37,38,39]. The third vaccine dose given more than three months after the primary vaccination has been shown in patients with RD to increase the RBD-IgG level, NAb titer against wt/Omicron and T cell response significantly, and compared to healthy controls lower or similar response levels have been reported [38,40]. In patients with CKD, the third dose has been demonstrated to increase the antibody levels, the proportion of participants with NAbs and NAb titer levels, and T cell responses [41,42,43,44], and seroconversion has been observed in patients who did not respond after two vaccine doses [35,45]. However, in kidney transplant recipients, the rates of response have remained suboptimal [42]. Due to the significant waning of antibody levels in the months following the third dose [39,46] and the emergence of SARS-CoV-2 variants with a high potential of immune escape, the European Center for Disease Prevention and Control and European Medicine Agency recommended, in July 2022, the administration of the fourth dose to people above 60 years of age as well as to vulnerable persons of any age [47]. The fourth vaccine dose has been shown to elicit higher antibody levels than the third dose in immunocompromised patients [48] but lower levels than in the controls in studies with patients on immunosuppressive therapy [49] and RD [50]. Few studies have, however, reported NAbs, T cell-mediated or hybrid immunity, and the persistence of antibodies after the fourth dose in patients with CKD and RD. One study reported increased neutralizing activity and T-cell response after the fourth dose in patients with RD. However, in only a minority of the subjects, the level of neutralization against the Omicron BA.2 strain was above baseline [50].

We report here significantly impaired antibody levels and NAb titers against the Omicron BA.1 strain after the fourth dose in participants with CKD and with OT. This finding suggests that even after a booster dose, the antibody-based immunity is limited in individuals living with CKD and even more limited in those living with CKD and OT. The progressive waning of antibody levels during the months after the fourth dose shown in this study and the emergence of SARS-CoV-2 variants with a high potential of immune escape highlight the need for booster vaccinations for people living with CKD. Despite lower humoral responses, vaccination can significantly reduce the odds of severe disease, hospitalization, and death in patients with CKD. However, for CKD patients with persistently poor vaccine responses, and especially for those with both CKD and OT, alternative preventive strategies such as new adapted vaccines or a combined approach of immunization, pre-exposure prophylaxis, and early post-exposure treatment using direct-acting antivirals and neutralizing monoclonal antibodies may be an option.

Previous data on antibody response to breakthrough infection in some particular immunocompromised patient groups, especially after the fourth vaccine dose, are scarce. Studies have reported higher antibody levels after hybrid compared to vaccine-induced immunity after two or three vaccine doses in patients with CKD [45,51] and immune-mediated inflammatory diseases [49]. We show, in this study, that despite impaired antibody responses to the fourth dose, most immunocompromised participants were able to attain a substantial antibody response to infection. At the time of the follow-up of our study (Feb-July 2022), molecular testing was not routinely performed in relation to infections with mild symptoms. Instead, at-home antigen tests were recommended. This practice is reflected in the relatively large proportion of participants (32%) whose breakthrough infection was detected solely via an antibody measurement. Due to the high overall attendance at the follow-up visit, it is unlikely that participants with a known or suspected infection were over-represented. During the follow-up, 7/109 (6%) of infections in immunocompromised participants and 6/19 (32%) in participants with CKD resulted in hospital admission. Our data demonstrate that also a relatively mild or asymptomatic infection can function as a booster in vaccinated immunocompromised individuals.

The strengths of our study include the evaluation of factors affecting immune response to the booster vaccination in immunocompromised participants with variable underlying conditions. Also, we compared immune responses to a group of healthy controls with matched vaccine products and a similar schedule for sampling. Because of the limited register data on the dose of the fourth mRNA-1273 vaccine (whole or half) and the greater immunogenicity of mRNA-1273 in comparison to BNT162b2 shown in this study, we performed the comparisons between disease groups after excluding participants who had received mRNA-1273. In addition, we assessed immune responses via antibody levels and NAbs against WT and the variant prevalent at the time of sampling, Omicron BA.1. Neutralizing antibodies are considered a more relevant correlate of protection than antibody levels against severe disease [52,53]. In addition to detecting previous SARS-CoV-2 infections via molecular tests, we also performed N-IgG antibody measurements and asked the participants about coronavirus at-home antigen tests to minimize the possibility of previous infections influencing our results.

Our study evaluated humoral immune responses to vaccination and infection, but we did not assess cellular immunity, which is a limitation in our study. Recent data show that despite impaired B-cell responses to vaccination, specific T-cell responses are less dependent on disease or treatment characteristics [38,40,50,54]. The ability to attain a T-cell response depends essentially on the extent to which immunodeficiency or immunosuppressive therapy affects T cells. Subjects with primary immune deficiencies were found to have robust T-cell responses which can confer protection against severe COVID-19 disease [54,55]. The register-based data used in our study was in part limited as not all details on immunosuppressive treatments were available, e.g., B-cell depleting therapies such as anti-CD20 or immunosuppressive drug mycophenolate mofetil which have been shown to negatively affect antibody responses [38]. We defined the study subgroups by the underlying disease (ICD-10 codes), notwithstanding the data from previous studies suggesting that the immune status, rather than the disease itself, is responsible for impaired immune responses [38,56]. Since most participants had more than one predisposing medical factor and the groups largely overlapped, a direct comparison between patient groups was not possible, which is also a limitation. During the ongoing follow-up of the study, we will assess T-cell responses and extend the register-based data.

The exposure of different population groups to infections may have varied, especially during the first COVID-19 epidemic wave, when containment measures were widely used. The proportion of immunocompromised participants who became infected (31%) during the first Omicron wave in this study was, however, quite similar to the age-standardized seroprevalence of the population (39%) in the Finnish population study carried out at the same time [16]. Considering that we excluded those participants who were infected before receiving the fourth vaccine dose, the cumulative seroprevalence in the immunocompromised participants would probably have been even closer to that of the general population.

The fourth dose of an mRNA vaccine given to immunocompetent individuals four months after the third dose was shown to restore antibody concentrations to the level seen after the third dose but not significantly boost the concentrations [57]. A longer interval between vaccine doses, or between vaccination and infection, is recognized to increase immunogenicity [58,59]. Very long intervals, up to at least 400 days, between vaccination and infection were shown to result in not only stronger antibody responses but also better cross-neutralization of different Omicron variants. In our study, the subjects had an infection after a relatively short time, within six months of the fourth vaccine dose. Despite this, the infection notably enhanced the antibody responses beyond the levels measured after the fourth dose. Further booster doses have been recommended for immunocompromised subjects, and some may have received up to eight doses by the end of 2023.

The long-term benefits of the continued boosting of immunocompromised individuals remain to be seen. In the current epidemiological situation, after many epidemic waves, most people have been infected and many even several times. In light of the current information, it should be critically evaluated whether booster doses are useful to be given repeatedly at short intervals to the immunocompromised or whether vaccinations should be scheduled closer to the start of the expected epidemic wave, e.g., in the autumn season.

## Figures and Tables

**Figure 1 vaccines-12-00247-f001:**
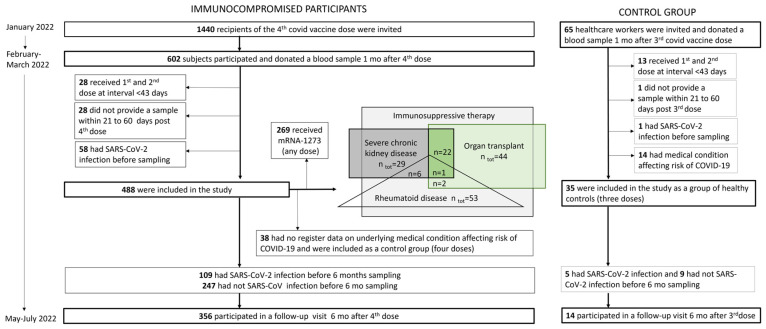
Study group flowchart showing the immunocompromised participants and the control groups included in the study.

**Figure 2 vaccines-12-00247-f002:**
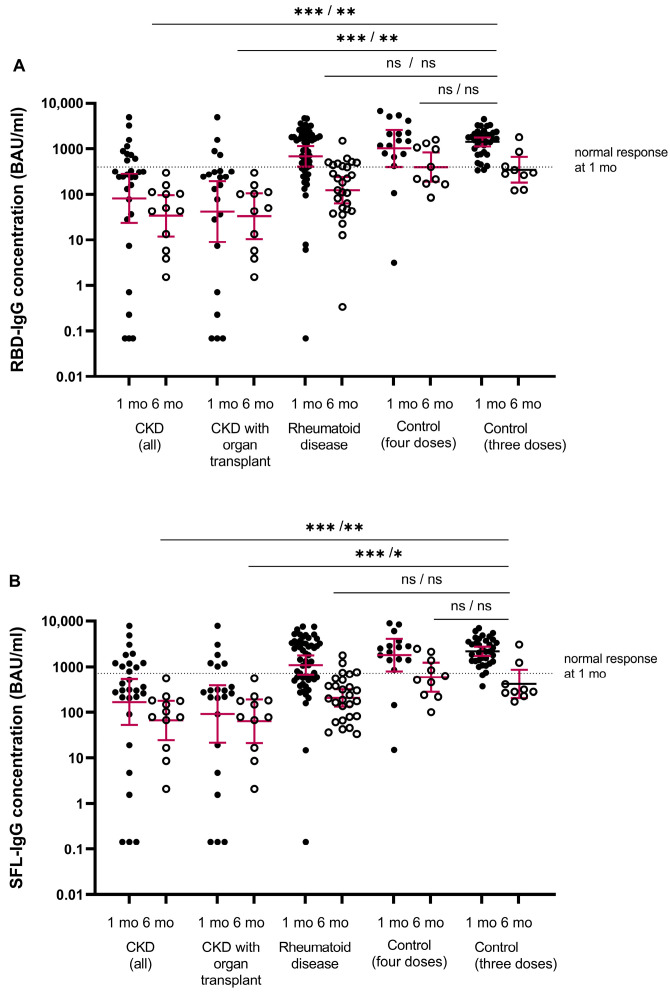
SARS-CoV-2 specific mean IgG antibody concentrations (GMC, 95% CI, binding antibody unit (BAU)/mL) to (**A**) receptor binding protein of spike protein (RBD-IgG) and (**B**) full-length spike protein (SFL-IgG). Data are shown for serum samples of participants with chronic kidney disease (CKD, all) (*n* = 29/13), CKD with organ transplant (*n* = 22/11), rheumatoid disease (*n* = 53/27) for controls after four doses (*n* = 17/10) and for controls after three doses (*n* = 35/9) at one/six months after the COVID-19 vaccine, respectively. Antibody levels at one and six months after the last vaccine are shown by closed and open circles, respectively. Participants who received the mRNA-1273 vaccine (any dose) are not included. The fifth percentile of RBD- and SFL-IgG concentrations determined for the control group one month after the third dose is shown as a threshold for a normal response (dashed line). Differences between each group and the control group were evaluated via the Mann–Whitney test. * *p* < 0.0025, ** *p* < 0.001, *** *p* < 0.0001, and ns = not significant.

**Figure 3 vaccines-12-00247-f003:**
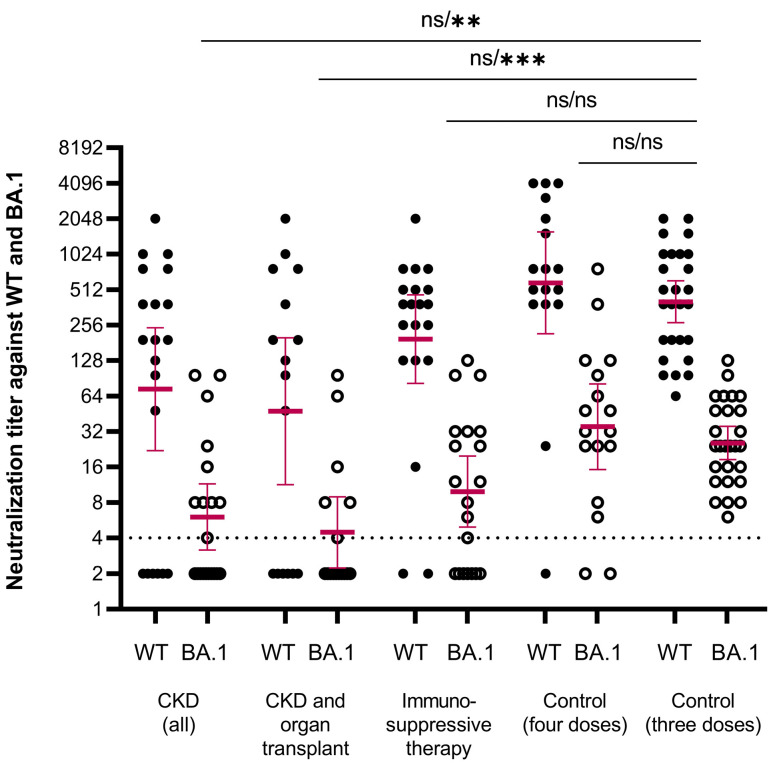
Neutralizing antibody titers against WT (closed circles) and BA.1 (open circles) SARS-CoV-2 strains. Data are shown for serum samples taken one month after the fourth dose of COVID-19 vaccine of participants with chronic kidney disease (CKD, all) (*n* = 20), CKD and organ transplant (*n* = 16), immunosuppressive therapy (*n* = 20), controls after four doses (*n* = 17) and controls after three doses (*n* = 27). Participants who received mRNA-1273 vaccine (any dose) are not included. The threshold for a positive titer (>4) is shown as a dashed line. Differences in NAb titers against WT and BA.1 strains between each group and the control group were evaluated by the Mann–Whitney test. ** *p* < 0.001, *** *p* < 0.0001, ns = not significant.

**Figure 4 vaccines-12-00247-f004:**
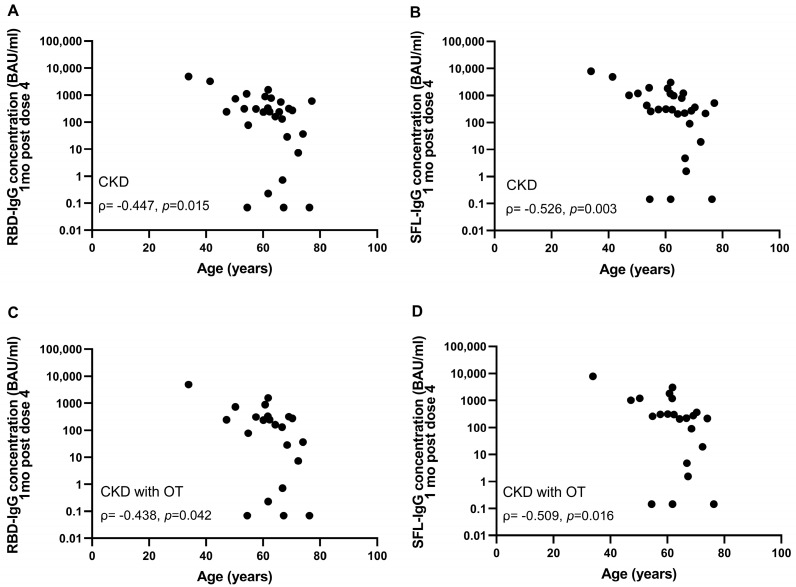
Correlation between age and RBD- (**A**,**C**) and SFL-IgG antibody concentrations (**B**,**D**) in CKD (*n* = 29) (**A**,**B**) and CKD with OT (*n* = 22) (**C**,**D**) groups one month after the fourth COVID-19 vaccine dose. Spearman correlation coefficients are shown.

**Figure 5 vaccines-12-00247-f005:**
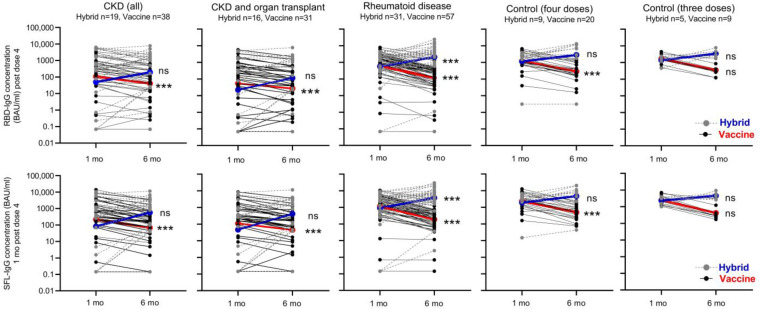
Kinetics of SFL- and RBD-IgG antibody concentrations (BAU/mL) in participants with hybrid or vaccine-induced immunity. Data are shown by disease and control groups at one and six months after booster dose of COVID-19 vaccine. Participants who became infected during the six-month follow-up and who thus developed hybrid immunity are shown by gray dotted lines/dots. Participants with vaccine-induced immunity are shown by black solid lines/dots. Geometric mean IgG concentrations of the groups are connected by blue (hybrid) and red (vaccine-induced immunity) lines. Significant differences in mean antibody concentrations between one and six months following the last vaccine dose in participants with vaccine-induced and hybrid immunity groups are shown; Wilcoxon signed-rank test, *** *p* < 0.0001, ns = not significant.

**Table 1 vaccines-12-00247-t001:** Baseline characteristics of study participants.

Demographics, Vaccination, and Infection Status	All Immunocompromised Participants		Chronic Kidney Disease, All		Chronic Kidney Disease and Organ Transplantation		Rheumatoid Disease		Controls (Four Vaccine Doses)		Controls (Three Vaccine Doses)	
Any COVID-19 vaccines *n*	488		85		66		122		38		35	
Age, years, median (IQR)	64.3	(52.1–69.4)	66.7	(60.2–71.0)	66.9	(61.6–70.6)	59.4	(45.4–67.0)	61.0	(47.9–67.6)	45.0	(36.5–55.6)
Female *n* (%)	285	(58.4)	42	(49.4)	31	(47.0)	89	(73.6)	27	(71.1)	30	(85.7)
Male *n* (%)	203	(41.6)	43	(50.6)	35	(53.0)	32	(26.4)	11	(28.9)	5	(14.3)
No mRNA-1273 vaccine *n* (%) *	198	(40.6)	29	(34.1)	22	(33.3)	53	(43.4)	17	(44.7)	35	(100)
Age, years, median (IQR)	62.8	(50.9–67.7)	62.3	(54.8–67.2)	62.1	(58.2–68.2)	55.3	(43.4–66.2)	63.5	(50.8–67.7)	45.0	(36.5–55.6)
Female *n* (%)	119	(60.1)	16	(55.2)	12	(54.5)	38	(71.1)	12	(70.6)	30	(85.7)
Male *n* (%)	79	(39.9)	13	(44.8)	10	(45.5)	15	(28.3)	5	(29.4)	5	(14.3)
No. of risk group classifications (median, IQR)	3	(2–4)	5	(4–6)	5	(4–6)	1	(1–2)	0	(0)	0	(0)
First vaccine *n* (%)												
BNT162B2	221	(45.3)	20	(23.5)	12	(33.3)	82	(67.8)	20	(52.6)	17	(48.6)
ChAdOx1 nCoV-19	245	(50.2)	61	(71.8)	51	(77.3)	32	(26.4)	16	(42.1)	18	(51.4)
mRNA-1273	22	(4.5)	4	(4.7)	3	(4.5)	7	(5.8)	2	(5.3)	0	(0)
Second vaccine *n* (%)												
BNT162B2	323	(66.2)	45	(52.9)	31	(47.0)	94	(77.7)	27	(71.1)	35	(100)
ChAdOx1 nCoV-19	123	(25.2)	29	(34.1)	26	(39.4)	18	(14.9)	8	(21.1)	0	(0)
mRNA-1273	42	(8.6)	11	(12.9)	9	(13.6)	9	(7.4)	3	(7.9)	0	(0)
Third vaccine *n* (%)												
BNT162B2	322	(66.0)	46	(54.1)	33	(50.0)	81	(66.9)	27	(71.1)	35	(100)
ChAdOx1 nCoV-19	0	(0)	0	(0)	0	(0)	0	(0)	0	(0)	0	(0)
mRNA-1273	166	(34.0)	39	(45.9)	33	(50.0)	40	(33.1)	11	(28.9)	0	(0)
Fourth vaccine *n* (%)												
BNT162B2	264	(54.1)	43	(50.6)	34	(51.5)	69	(57.0)	21	(55.3)	0	(0)
ChAdOx1 nCoV-19	0	(0)	0	(0)	0	(0)	0	(0)	0	(0)	0	(0)
mRNA-1273	224	(45.9)	42	(49.4)	32	(48.5)	52	(43.0)	17	(44.7)	0	(0)
COVID-19 ^#^ (hybrid immunity) *n*/*n* ^§^ (%)	109/356	(30.6)	19/57	(33.3)	16/47	(34.0)	31/88	(34.9)	9/29	(31.0)	5/14	(36)

Of all immunocompromised participants selected to this study (*n* = 488), baseline characteristics of three subgroups subjected to further analyses of antibody responses are shown. * Percentage of participants who did not receive mRNA-1273 of all participants within the group. ^#^ COVID-19 after four/three doses of vaccine (hybrid immunity). ^§^ Samples available at follow-up sampling six months after last vaccine dose.

**Table 2 vaccines-12-00247-t002:** Posterior means and 95% credible intervals (95% CI) for regression coefficients from the Bayesian gamma regression model explaining IgG antibody concentrations specific for full-length spike protein (SFL) and receptor binding domain of spike protein (RBD).

Variable	N	Estimate (SFL-IgG)	95% CI (SFL-IgG)	Estimate (RBD-IgG)	95% CI (RBD-IgG)
Age (in years)		0.995	(0.985–1.003)	0.995	(0.984–1.003)
Sex—female		1.001	(0.874–1.169)	1.001	(0.876–1.159)
Vaccine—mRNA-1273 (fourth dose)	224	1.568	(1.161–2.018)	1.510	(1.092–1.968)
Cancer under treatment	149	1.088	(0.954–1.534)	1.098	(0.963–1.606)
Autoimmune diseases *	203	0.997	(0.808–1.129)	0.999	(0.833–1.149)
Organ or stem cell transplant	113	0.993	(0.764–1.124)	0.994	(0.768–1.142)
Neurological disease or condition that interferes with breathing	27	1.009	(0.859–1.463)	1.013	(0.879–1.614)
Immunosuppressive therapy	340	0.612	(0.455–0.845)	0.652	(0.482–0.940)
Type 2 diabetes with drug therapy	79	1.004	(0.877–1.293)	1.002	(0.856–1.264)
Type 1 diabetes or adrenal insufficiency	46	0.895	(0.491–1.059)	0.878	(0.474–1.064)
Sleep apnea	68	1.032	(0.926–1.541)	1.021	(0.923–1.435)
Severe chronic pulmonary disease	63	1.000	(0.829–1.185)	0.999	(0.817–1.196)
Severe chronic liver disease	19	1.001	(0.798–1.393)	1.006	(0.838–1.498)
Severe chronic kidney disease	85	0.479	(0.339–0.741)	0.490	(0.342–0.787)
Severe disorders of the immune system	21	0.991	(0.631–1.177)	0.994	(0.633–1.222)
Severe heart diseases e.g heart failure	166	0.990	(0.771–1.127)	0.989	(0.754–1.108)

* Including diseases/conditions where immunosuppressive therapy is generally used (e.g., rheumatoid diseases, psoriasis, and inflammatory bowel disease).

**Table 3 vaccines-12-00247-t003:** Percentages of seropositive participants, participants with a normal S-IgG response and with NAbs, and participants with RBD-IgG concentration > 200 BAU/mL one month after the booster dose.

	S-IgG % Seropositive (*n*/*n*) at 1 mo		S-IgG % with Normal Response (*n*/*n*) at 1 mo		NAb % Positive (*n*/*n*) WT at 1 mo		NAb % Positive (*n*/*n*) BA.1 at 1 mo		RBD-IgG >200 BAU/mL % (*n*/*n*) at 1 mo	
CKD (all)	83	(24/29)	31 ***	(9/29)	70 *	(14/20)	45 ***	(9/20)	45 ***	(13/29)
CKD with organ transplant	77	(17/22)	18 ***	(4/22)	63 *	(10/16)	38 ***	(6/16)	50 ***	(11/22)
Immunosuppressive therapy/ rheumatoid disease ^#^	98	(52/53)	64 **	(34/53)	90	(18/20)	60 **	(12/20)		
					100	(10/10)	50 **	(5/10)	87	(46/53)
Control (four doses)	100	(17/17)	88	(15/17)	94	(16/17)	88	(15/17)	88	(15/17)
Control (three doses)	100	(35/35)	94	(33/35)	100	(27/27)	100	(27/27)	100	(35/35)

Significant differences vs. control group are shown; Fischer’s exact test, * *p* < 0.004, ** *p* < 0.001, *** *p* < 0.0001. ^#^ Of the participants on immunosuppressive therapy (IT), 53/112 had rheumatoid disease (RD). NAbs were determined for 20/112 randomly selected participants with IT, and 10/20 of them also had RD.

## Data Availability

The data presented in this study are available, upon a reasonable request, from the corresponding author.

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
