# Peer review of "Hybrid Immunity Improves the Immune Response after the Fourth COVID-19 Vaccine Dose in Individuals with Medical Conditions Predisposing to Severe COVID-19"

_vaccines, 2024, doi:10.3390/vaccines12030247_

Round 1
Reviewer 1 Report
Comments and Suggestions for Authors
Ekström et al. examine SARS-CoV-2 antibody responses (spike-IgG and neutralizing; for the ancestral and Omicron BA.1 strains) after the fourth vaccine dose and subsequent infection, to define patient groups benefiting most from vaccine boosters. Also healthy controls are included. This piece of work appears skillfully planned, executed and written, and lacks major shortcomings. The pdf version attached contains for the authors' consideration, a number of _minute_ comments or suggestions (in highlightings or sticky notes), for their optional fine-tuning of the text.

Comments on the Quality of English LanguageThe attachment contains a number of minute comments from me, for the authors' optional consideration.
Author Response
Dear Reviewer and Editor,
Thank you for considering our manuscript for publication. We thank for positive feedback and important comments which significantly improved the manuscript. We have changed the manuscript in accordance with the comments and replied below.
Sincerely,
Nina Ekström
Finnish Institute for Health and Welfare
Helsinki, Finland
Open Review #1
Comments and Suggestions for Authors
The pdf version attached contains for the authors' consideration, a number of _minute_ comments or suggestions (in highlightings or sticky notes), for their optional fine-tuning of the text.
Reply: Thank you, we have edited the text based on the suggestions.
Reviewer 2 Report
Comments and Suggestions for Authors
Reviewer’s Comments - Manuscript Vaccines-2834514 “Hybrid immunity improves the immune response after the fourth Covid-19 vaccine dose in individuals with medical conditions predisposing to severe Covid-19” by N. Ekström et al.
In the manuscript the Authors evaluated the spike protein-specific IgG and neutralizing antibodies against the ancestral and Omicron BA.1 strains one month after the fourth dose from 488 adult immunocompromised participants and compared to the levels of healthy controls after three doses of vaccines. They used Bayesian generalized linear modeling to assess factors explaining antibody concentrations and assessed vaccine-induced and hybrid immunity six months after the last vaccine dose. Lower antibody responses and Nab were observed in subjects with chronic kidney disease and immunosuppressive therapy. By the 6 months after the last dose, one third of the participants became infected, which enhanced antibody levels in most immunocompromised participants. Authors suggest that impaired antibody responses, especially Nabs against the Omicron lineage, predict limited protection in individuals with CKD and highlight the need for alternative pharmaceutical preventive strategies.
The study fits the scope of the journal and the topic will be of interest to the readership of Vaccines Journal and more specifically to researchers working in the field of Sars-CoV-2 research and vaccine development. However, in my opinion, the Authors should improve the presentation of their results and clarify few points before the manuscript can be accepted for publication.
-As a major comment, I think the organization of the manuscript needs to be revised to better clarify the study design and results to the reader. I feel that the study design and all the cohorts including the controls need to be better described in the main text and in figure 1/Tables (i.e. in the Materials and Methods section 2.1 and in supplementary information). Somewhere I found it confusing to follow the text and the figures/tables with regard to the study groups and their numbers. Please also define/clarify better the risk groups and the controls and the differences between non risk group vs controls and their numbers. In figure 1, I suggest to insert a timeline schedule and the controls. In figure 1 (and text) it is not clear to me if all 443 participants were tested, including the 84 who received the 5th dose before the 6 mo sampling. If this is the case, why they were tested after the 5th dose?
-Could you clarify why the healthy controls were tested after the third dose and not after fourth dose?
- legend to figures and tables: I suggest to describe in the caption just the figure/table content, any other information should be in the text. Also insert a proper title in figures and tables.
- line 188: what do you mean for “normal” antibody concentration? Perhaps the range of antibody levels in normal subjects could be indicated in parenthesis.
- lines 209-212: if sampling were done (based on the study design) one and six months after the fourth dose, why participants would have received a Covid vaccines between the samplings? It is not clear the sentence “Of the participants who had not received Covid-19 vaccines between the samplings (n=356), 109 (31 %) experienced….. (Table 1)”. Could you please clarify this point?
-lines 213-220: it would be better to insert a table (i.e. in supplementary material) describing the participants (i.e. how many had 0, 1, 2, 3 or more predisposing conditions) and reporting the data described in this paragraph.
-Table 1: the numbers are not clear to me: i.e. if the total number of participants is 488 (Column All immunocompromised participants), why the sum of the participants of the other columns (85, 66, 122, 38, 35) is 346?
Perhaps “%” is missing after “Any Covid-19 vaccines n%. Also it looks like that a Table caption is missing regarding the last Table line (Covid-19 (hybrid immunity) n/n§ (%): what does the symbol § mean?
-Section 3.2: how many healthy controls were included? 65 (as stated at line 105) or 35 as stated here? As said above, I think that the description of the participants including the controls should be done mainly in one section (i.e. materials and methods and supplementary information).
-lines 255-258: in figure S1 the group with OT and no CKD are reported. So why you state that “We did not asses the group with OT (without CDK….)? Please clarify this point.
-lines 299-301: can you show the data in a figure?
Table 3: a Table caption is missing for #/ after Immunosuppressive therapy
Is % of infected subjects significantly different between of controls (36%) and immunosuppressed (31%)? Also there were important differences in disease symptoms between controls and immunosuppressed?
Author Response
Dear Reviewer and Editor,
Thank you for considering our manuscript for publication. We thank for positive feedback and important comments which significantly improved the manuscript. We have changed the manuscript in accordance with the comments and replied to questions below:
Sincerely,
Nina Ekström
Finnish Institute for Health and Welfare
Helsinki, Finland
Open Review#2
Comments and Suggestions for Authors
In the manuscript the Authors evaluated the spike protein-specific IgG and neutralizing antibodies against the ancestral and Omicron BA.1 strains one month after the fourth dose from 488 adult immunocompromised participants and compared to the levels of healthy controls after three doses of vaccines. They used Bayesian generalized linear modeling to assess factors explaining antibody concentrations and assessed vaccine-induced and hybrid immunity six months after the last vaccine dose. Lower antibody responses and Nab were observed in subjects with chronic kidney disease and immunosuppressive therapy. By the 6 months after the last dose, one third of the participants became infected, which enhanced antibody levels in most immunocompromised participants. Authors suggest that impaired antibody responses, especially Nabs against the Omicron lineage, predict limited protection in individuals with CKD and highlight the need for alternative pharmaceutical preventive strategies.
The study fits the scope of the journal and the topic will be of interest to the readership of Vaccines Journal and more specifically to researchers working in the field of Sars-CoV-2 research and vaccine development. However, in my opinion, the Authors should improve the presentation of their results and clarify few points before the manuscript can be accepted for publication.
-As a major comment, I think the organization of the manuscript needs to be revised to better clarify the study design and results to the reader. I feel that the study design and all the cohorts including the controls need to be better described in the main text and in figure 1/Tables (i.e. in the Materials and Methods section 2.1 and in supplementary information). Somewhere I found it confusing to follow the text and the figures/tables with regard to the study groups and their numbers. Please also define/clarify better the risk groups and the controls and the differences between non risk group vs controls and their numbers. In figure 1, I suggest to insert a timeline schedule and the controls.
Reply: We have done some reorganization of the manuscript and have described the cohorts and controls in the main text and figures/tables more clearly. We have also described the differences between non risk group vs controls and their numbers more clearly, in main text and figures. We have made major revisions to Figure1 and have included the controls and inserted a timeline into the figure.
In figure 1 (and text) it is not clear to me if all 443 participants were tested, including the 84 who received the 5th dose before the 6 mo sampling. If this is the case, why they were tested after the 5th dose?
Reply: All participants were tested. Since antibody response to the 5th dose was not within the scope of this manuscript, we have clarified the presentation of study design and methods as well as Figure 1 by excluding this group of participants. The original aim of the study was to assess antibody levels one and six months after the fourth dose in all immunocompromised participants. However, since the fifth vaccine dose was in Finland recommended for immunocompromised individuals during the follow-up of the study, in June 2022, some of the participants received the fifth dose before the follow-up sampling. All participants received Covid-19 vaccines in accordance with the national recommendations. The participants were invited by mail and were instructed to donate a blood sample at their local healthcare district laboratory. Therefore, it was unfortunately not always possible to schedule the 6-month sampling before 5th vaccine dose.
-Could you clarify why the healthy controls were tested after the third dose and not after fourth dose?
Reply: All study participants received Covid-19 vaccines in accordance with the national recommendations. In Finland, the fourth Covid-19 vaccine doses have been to date recommended only for certain risk groups. For non-risk group adults <65 years, including healthcare personnel, three Covid-19 vaccine doses are recommended. In this study, we compared antibody levels after the booster dose i.e the fourth dose for the immunocompromised and the third dose for the healthy controls. To assess antibody levels after three vs four doses of vaccine, we included an additional control group of participants who had received four vaccine doses but had no predisposing medical factors. The reason for administration of the fourth dose in this small group was probably ambiguities regarding the specific risk groups when the recommendations were initially given.
- legend to figures and tables: I suggest to describe in the caption just the figure/table content, any other information should be in the text. Also insert a proper title in figures and tables.
Reply: We have edited the legends in the figures and tables as the reviewer suggested.
- line 188: what do you mean for “normal” antibody concentration? Perhaps the range of antibody levels in normal subjects could be indicated in parenthesis.
Reply: We assigned the lower limit of a normal antibody response to vaccination at the 5th percentile of the SFL- and RBD-IgG antibody concentrations in the group of healthy controls (after three doses of vaccine). We have added this lower limit of normal antibody level in parenthesis in the text, as suggested.
- lines 209-212: if sampling were done (based on the study design) one and six months after the fourth dose, why participants would have received a Covid vaccines between the samplings? It is not clear the sentence “Of the participants who had not received Covid-19 vaccines between the samplings (n=356), 109 (31 %) experienced….. (Table 1)”. Could you please clarify this point?
Reply: The original aim of the study was to assess antibody levels one and six months after the fourth dose in all immunocompromised participants. However, since the fifth vaccine dose was in Finland recommended for immunocompromised individuals during the follow-up of the study, in June 2022, some of the participants received the fifth dose before the follow-up sampling. All participants received Covid-19 vaccines in accordance with the national recommendations. The participants were invited by mail and were instructed to donate a blood sample at their local healthcare district laboratory. Therefore, it was unfortunately not always possible to schedule the 6-month sampling before 5th vaccine dose. To clarify the description of the study design, study groups and methods in the text and Figure 1, we have excluded the description of the group of participants who received five doses.
-lines 213-220: it would be better to insert a table (i.e. in supplementary material) describing the participants (i.e. how many had 0, 1, 2, 3 or more predisposing conditions) and reporting the data described in this paragraph.
Reply: We have added a table describing the participants in the supplementary material, as the reviewer suggested.
-Table 1: the numbers are not clear to me: i.e. if the total number of participants is 488 (Column All immunocompromised participants), why the sum of the participants of the other columns (85, 66, 122, 38, 35) is 346?
Reply: The total number of immunocompromised participants is 488. We focused further evaluation of antibody responses to those immunocompromised subgroups we had identified by modeling i.e individuals receiving immunosuppressive therapy and chronic kidney disease (CKD). We selected (as described in the Results) participants with rheumatoid diseases (n=122) to represent participants receiving immunosuppressive therapy. All individuals living with CKD are evaluated as one group (n=85), and we additionally evaluated separately those individuals living with CKD and organ transplant (n=66). Not all immunocompromised participants belong to these subgroups, and therefore the sum of individuals in the subgroups in Table 1 does not add to 488. We have tried to clarify this in the Table legend. We have also tried to clarify the two control groups by editing the description of the control groups in Table 1.
Perhaps “%” is missing after “Any Covid-19 vaccines n%. Also it looks like that a Table caption is missing regarding the last Table line (Covid-19 (hybrid immunity) n/n§ (%): what does the symbol § mean?
Reply: “Any Covid-19 vaccines n” should be as it is i.e the number of individuals in each group. The caption was missing by mistake, but we have now added it. Thank you for noticing.
-Section 3.2: how many healthy controls were included? 65 (as stated at line 105) or 35 as stated here? As said above, I think that the description of the participants including the controls should be done mainly in one section (i.e. materials and methods and supplementary information).
Reply: We have reorganised the text and the description of the participants is now mainly given in materials and methods section. Section 3.2 has been removed from the Results. We have clarified the description of the control group in the materials and methods section and in Figure 1.
-lines 255-258: in figure S1 the group with OT and no CKD are reported. So why you state that “We did not asses the group with OT (without CDK….)? Please clarify this point.
Reply: This is correct. We have rephrased this sentence.
-lines 299-301: can you show the data in a figure?
Reply: We have added a new figure (Figure 4) which illustrates correlation between age and antibody concentration in the CKD and CKD with OT groups.
Table 3: a Table caption is missing for #/ after Immunosuppressive therapy
Reply: We have excluded “#” as unnecessary. All necessary information is given in the figure legend.
Is % of infected subjects significantly different between of controls (36%) and immunosuppressed (31%)? Also there were important differences in disease symptoms between controls and immunosuppressed?
Reply: These are interesting questions. The percentage of individuals who experienced an infection during the follow-up was statistically not different between these groups. We have added this information in section 3.5. We did not collect information on disease symptoms and can therefore not answer the latter question. We did, however, evaluate periods of hospitalization at the time of the laboratory-confirmed SARS-CoV-2 infection in the immunocompromised and control groups. The numbers of hospitalization periods within groups are described in the Results and have been discussed. Because of the very low number of participants with infection in the control group, it was not reasonable to assess differences between groups by statistical tests.
Round 2
Reviewer 2 Report
Comments and Suggestions for Authors
Thank you. I acknowledge the revision of the manuscript and its improvement.
I still suggest however some minor revisions, i.e. for each Table it would be preferable to have a title, whereas the descption in my opinion would be below the table or in the text
Author Response
Dear Reviewer,
We thank for positive feedback and for your persistence. We have now revised the manuscript in accordance with your suggestion. We have given Tables 1-3 shorter titles and have inserted all complementary description below the tables or in the main text. We have marked the changes made to the manuscript in the 2nd round in yellow.
Sincerely,
Nina Ekström
Finnish Institute for Health and Welfare
Helsinki, Finland